# Impaired cellular energy metabolism in cord blood macrophages contributes to abortive response toward inflammatory threats

Stephan Dreschers[1], Kim Ohl[1], Michael Lehrke[2], Julia Möllmann[2], Bernd Denecke[3], Ivan Costa[3], Thomas Vogl[4], Dorothee Viemann[5], Johannes Roth[4], Thorsten Orlikowsky[1] & Klaus Tenbrock[1]

Neonatal sepsis is characterized by hyperinflammation causing enhanced morbidity and mortality compared to adults. This suggests differences in the response towards invading threats. Here we investigate activated cord blood macrophages (CBMΦ) in comparison to adult macrophages (PBMΦ), indicating incomplete interferon gamma (IFN-γ) and interleukin 10 (IL-10)-induced activation of CBMΦ. CBMΦ show reduced expression of phagocytosis receptors and cytokine expression in addition to altered energy metabolism. In particular, IFN-γ as well as IL-10-activated CBMΦ completely fail to increase glycolysis and furthermore show reduced activation of the mTOR pathway, which is important for survival in sepsis. MTOR inhibition by rapamycin equalizes cytokine production in CBMΦ and PBMΦ. Finally, incubation of PBMΦ with cord blood serum or S100A8/A9, which is highly expressed in neonates, suppresses mTOR activation, prevents glycolysis and the expression of an PBMΦ phenotype. Thus, a metabolic alteration is apparent in CBMΦ, which might be dependent on S100A8/A9 expression.

[1] Department of Pediatrics, RWTH Aachen University, Aachen, Germany. [2] Department of Medicine, RWTH Aachen University, Aachen, Germany. [3] Computational Biology, IZKF, RWTH Aachen University, Aachen, Germany. [4] Institute of Immunology, University of Münster, Münster, Germany. [5] Experimental Neonatology, Medical University of Hannover (MHH), Münster, Germany. These authors contributed equally: Stephan Dreschers, Kim Ohl, Klaus Tenbrock, Thorsten Orlikowsky. Correspondence and requests for materials should be addressed to T.O. (email: torlikowsky@ukaachen.de)

Sepsis is still the leading cause of death in neonates in under-resourced countries. The risk of sepsis is particularly high in neonates compared with older children and adults; however, the reasons behind it are incompletely understood. Neonatal sepsis is predominately caused by Gram + group B streptococci and causes a variety of sequelae with inflammatory organ damage and is accompanied by impaired apoptotic depletion of monocytes, aberrant cytokine production, and sustained inflammation.

Macrophages (MΦ) are at the front line of the innate immune system and differentiate from monocytes after infiltrating the infected tissue. Depending on the local cytokine milieu and specific microenvironment cues, macrophages have different activation types with different functions[1,2]. MΦ have traditionally been classified into two categories: type 1 or classically activated MΦ (M1-MΦ), and type 2 or alternatively activated MΦ (M2-MΦ). Under physiological conditions, monocytes are constantly exposed to M-CSF, which induces substantial proliferation combined with gene activation in vivo[3]. Administration of IFN-γ or LPS rearranges the transcriptome of these M-CSF-induced MΦ dramatically. Compared with M-CSF, MΦ, IFN-γ, or LPS-induced M1−MΦ differ in 90% of active genes[3]. Furthermore, M-CSF MΦ can be alternatively differentiated by administration, e.g., of IL-4, IL-13, or IL-10. In contrast to IFN-γ or LPS-induced MΦ, they are characterized by an enhanced phagocytic capability, which is functionally shaped to eliminate cellular and apoptotic debris rather than to neutralize pathogens[4]. In addition, their APC functions are reduced[5].

Intense transcriptional reprogramming during polarization includes activation of distinct metabolic pathways. Therefore, the activation phenotype is tightly linked to metabolism, which is not only needed for energy metabolism, but moreover, also involved in transcriptional regulation of macrophages[1,2]. LPS induces a metabolic shift from oxidative phosphorylation to glycolysis in macrophages[2] and this LPS-dependent upregulation is partially dependent on HIF1α[6]. The activation of glycolysis pathways results in rapid abundance of adenosine triphosphate (ATP), which is important in the context of acute bacterial infection because glycolysis, although less efficient in generating ATP, can be upregulated many fold and therefore results in a faster production of ATP compared with oxidative phosphorylation[7]. Thus, it is suggested that increased utilization of glucose is associated with a proinflammatory macrophage phenotype[2,6]. In contrast, in IL-4-activated MΦ oxidative phosphorylation is the primary and more efficient way of energy supplementation[8,9]. The metabolic signaling pathways activated by polarizing signals include, among others, AKT, mTORC1, mTORC2, and AMPK depending on the stimuli used[10].

Nevertheless, most of what we know about metabolic programs in macrophages originates from studies in the murine system and there is increasing evidence that human macrophage biology differs from that of commonly studied murine models[11]. It is of note that, despite their importance in the primary immune defense, an unbiased comprehensive transcriptional, epigenetic, and metabolic analysis of neonatal macrophages during differentiation is lacking until now. This is surprising, since a high risk of neonatal death from sepsis might result from altered responses by innate immune cells. It was recently shown by the group of Viemann that the alarmins S100A8 and S100A9 are not only abundant in high amounts, but moreover, play a central role in the control of inflammatory monocytes in neonates and prevent harmful hyperinflammation by inducing a selective, transient microbial unresponsiveness[12,13].

In this study, we now show that cord blood-derived and in vitro-activated macrophages reveal transcriptional and metabolic alterations, in particular, regarding defective glycolysis, which affects polarization and function. These alterations are mTOR dependent and can be partially induced in adult macrophages by incubation with either cord blood or S100A8/A9 during the differentiation process.

## Results

### Polarized CBMΦ reveals marked alterations compared with PBMΦ.
Macrophages are activated by various signals to acquire specialized functions; however, macrophage phenotypes exhibit a high functional plasticity and diverse mediators have been used to activate and polarize macrophages. To test the activation potential of neonatal cord blood monocytes and adult peripheral monocytes, we activated M-CSF-induced macrophages with IL-10 and IFN-γ, respectively, to achieve MΦ(IL-10) and MΦ(IFN-γ) and used the nomenclature linked to the activation standards, as suggested by Murray et al.[14]. We decided to use IL-10 and IFN-γ as activating cytokines, since both are highly upregulated during neonatal sepsis[15] and the typical pathogen of neonates are Gram+ group B streptococci, which do not have LPS that is classically used to differentiate macrophages.

To investigate macrophage polarization on neonatal and adult monocytes, we stimulated blood-derived mononuclear cells from cord blood and from peripheral blood from adults with M-CSF for 3 days and added IFN-γ to achieve classical activated MΦ (IFN-γ)-polarized macrophages and IL-10 for an additional 48 h to polarize macrophages toward an alternatively activated MΦ (IL-10) phenotype. In comparison with adult undifferentiated macrophages (PBMΦ(0)), cord blood-derived macrophages (CBMΦ(0)) revealed an altered expression of several surface markers, including CD14, HLA-DR (tendencially), CD16, CD32, CD64, and CD163, but expressed the comparable levels of IFN-γ and IL-10 receptors (Fig. 1a, b, gating strategies are shown in Supplementary Fig. 1). After polarization with IFN-γ, M1 markers like CD14, HLA-DR, CD80, and CD86 were much less expressed on cord blood macrophages (Fig. 1c). Furthermore, after polarization with IL-10, typical M2 markers CD16, CD32, CD64, and the scavenger receptor CD163 were not expressed properly in cord blood macrophages; furthermore, arginase expression and STAT3 phosphorylation occurred at lower levels (Fig. 1d). These data suggest that cord blood macrophages display broad alterations during activation compared with adult macrophages, which are irrespective of the differentiation process.

### CBMΦ shows an altered transcriptional regulation compared with PBMΦ.
Polarizing signals upregulate transcriptional programs to enforce macrophage activation. We therefore performed whole- transcriptome analysis to uncover transcriptional differences, which might account for phenotypical alterations of cord blood macrophages and observed a remarkable different transcriptional profile of cord blood-derived activated macrophages compared with adult-activated macrophages (Fig. 2a). Interestingly, we noticed an altered expression of several genes, which belong to metabolic pathways (Fig. 2b, c). Macrophages from cord blood showed a downregulation of several genes that are involved in glucose metabolism (Fig. 2d). In detail, cord blood-derived MΦ(IFN-γ) macrophages showed a significantly reduced expression of glucose transporter 1 (Glut1) (Fig. 2e), which is the primary rate-limiting glucose transporter on proinflammatory-polarized MΦs[2]. MΦ(IL-10) as well as MΦ(IFN-γ) macrophages both revealed reduced expression of phosphofructokinase M (PFKM) (Fig. 2f), which catalyzes the first committed step of glycolysis by the phosphorylation of D-fructose 6-phosphate to fructose 1,6-bisphosphate. Glycolysis-derived pyruvate can be imported into mitochondria via the mitochondrial pyruvate carrier (MPC), the subunits MPC1 and MPC2 of which were downregulated in MΦ(IL-10) macrophages (Fig. 2g, h). Furthermore, pyruvate carboxylase (PC) and malate dehydrogenase 2

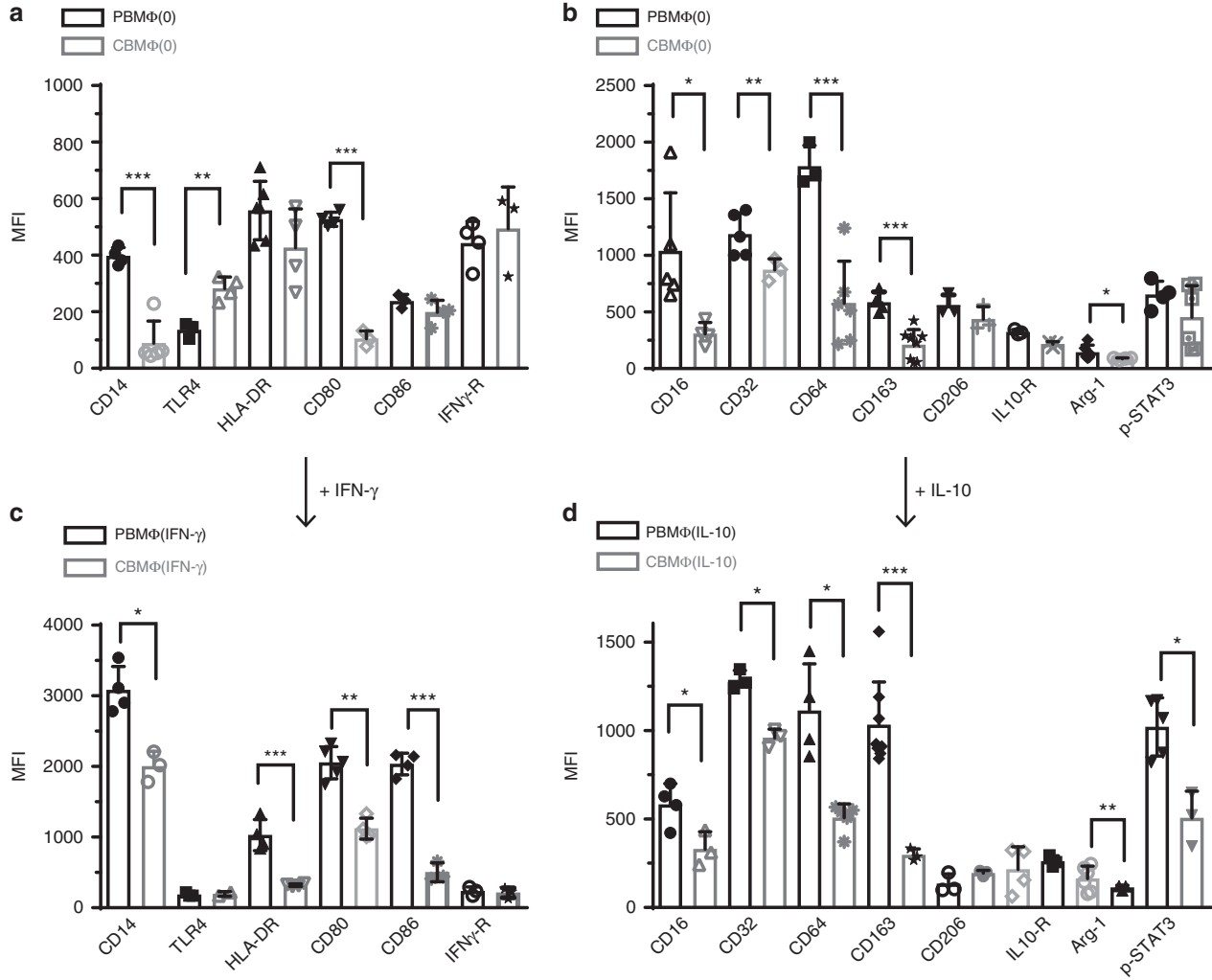

**Fig. 1** Polarization toward MΦ(IFN-γ) and MΦ(IL-10) states is altered in cord blood macrophages. **a**, **b** Polarization was induced in PBMC (peripheral blood-derived mononuclear cells, black-framed bars) and CBMC (cord blood-derived mononuclear cells, gray-framed bars) using 100 ng/ml hM-CSF for 72 h in RPMI (upper charts, MΦ(0)). Further polarization was achieved by addition of 50 ng/ml IFN-γ (designated as MΦ(IFN-γ)) (**c**) or 10 ng of IL-10 (designated as MΦ(IL-10)) (**d**) for an additional 48 h. Immunotyping was performed by flow cytometric analysis detecting selected surface markers of traditional M1–MΦ (left charts) and traditional M2–MΦ (right charts). MFI was determined as a mean value; represented data show MFI of a marker—MFI of isotype control. Significance of reduced expression in CBMΦ was tested by utilizing the ANOVA test followed by Bonferroni post test ($N = 3$–8, *$p < 0.05$; **$p < 0.01$; ***$p < 0.001$). Bars indicate mean and error bars SD (detailed statistics in supplemental informations)

(*MDH2*) which are not only involved in gluconeogenesis but also play an anaplerotic role for the TCA also showed a reduced expression in both MΦ(IL-10) as well as MΦ(IFN-γ) cord blood-derived macrophages (Fig. 2i, j).

**CBMΦ reveals broad defects in energy metabolism.** Activated macrophages undergo profound reprogramming of their cellular metabolism. Our whole-transcriptome analysis suggested a defect in activation of metabolic pathways in cord blood-derived macrophages. We therefore analyzed adult and cord blood macrophages for changes in the rate of extracellular acidification (ECAR) and the mitochondrial rate of oxidation (OCR) as a measure for glycolysis and OXPHOS, respectively. We noticed a broad metabolic defect in polarized cord blood macrophages, which, in particular, affected metabolization of glucose. A defect in anaerobic glucose metabolism was apparent in all polarized cord blood macrophages (Fig. 3a, b), while oxidative consumption rate was only affected in MΦ(IFN-γ) (Fig. 3c, d). This is quite striking, since these macrophages are in the first line of defense against invading pathogens. In particular, a shift to glycolysis,

which is shown to be altered in cord blood-derived MΦ(IL-10) and in MΦ(IFN-γ) macrophages, is associated with acute bacterial infections and attenuation of glycolysis reduces inflammatory cytokine production and bacterial killing[16]. Interestingly, ECAR in MΦ(IFN-γ) is below the rate of MΦ(IL-10) in our hands; however, this is in line with a very recent paper of Wang et al.[17], which shows an induction of glycolysis of IFN-γ-induced macrophages, however, at clearly lower levels compared with LPS-induced macrophages. To the best of our knowledge, plainly IL-10-induced macrophages have not been investigated yet with regard to metabolism in the human system.

**mTOR activation is reduced in CBMΦ.** mTOR and AKT are major metabolic regulators and are critically involved in the control of macrophage metabolism and activation[18]. Our findings that cord blood-derived macrophages reveal metabolic defects led us to consider a role for mTOR in this process. Indeed, we found that the total mTOR protein expression (Fig. 4b) as well as phosphorylation (Fig. 4a, c) were downregulated in cord blood-derived MΦ(0), MΦ(IFN-γ), and MΦ(IL-10) macrophages

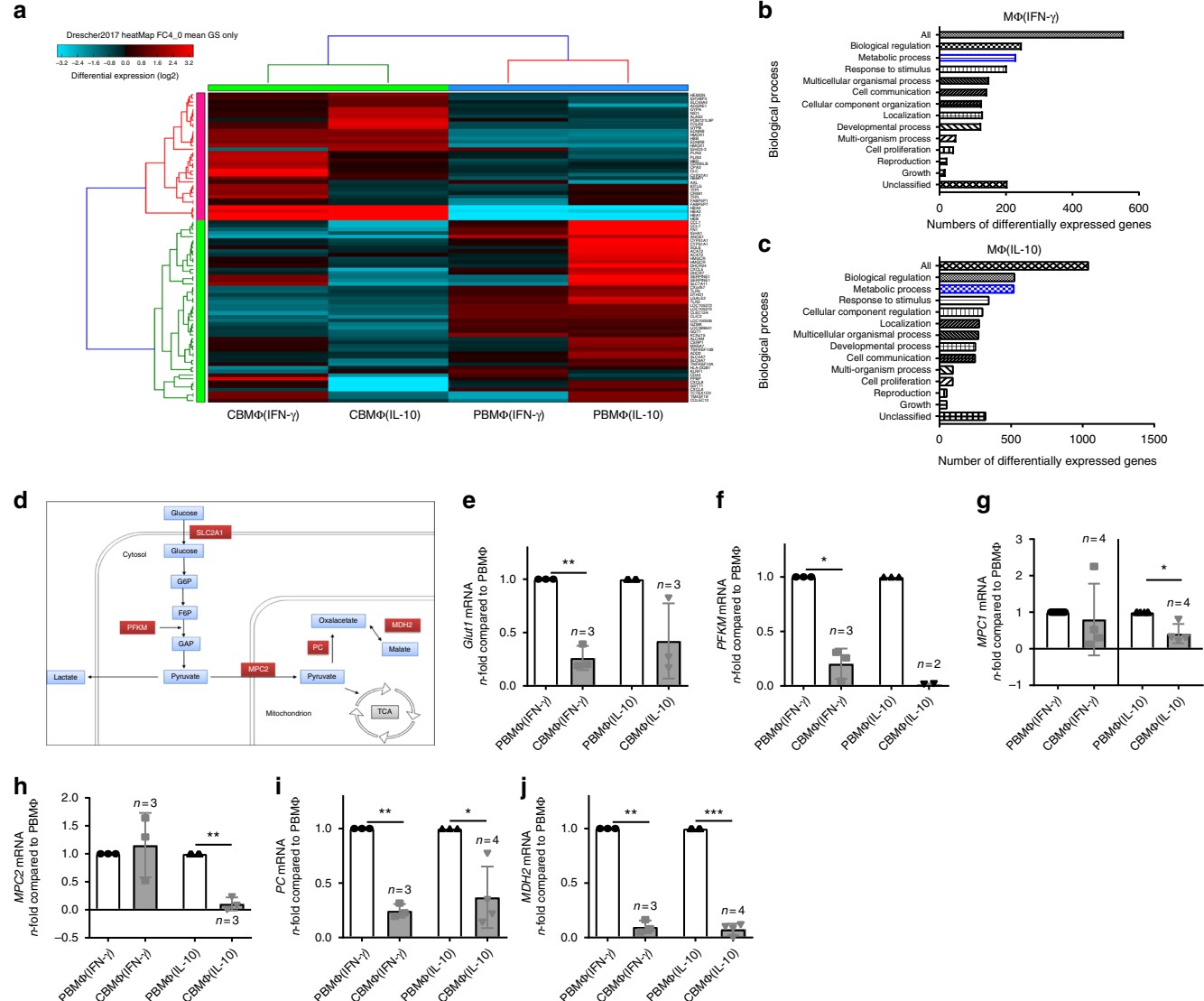

**Fig. 2** Whole-transcriptome analysis reveals broad metabolic aberrations in polarized cord blood macrophages. **a** Heatmap of differently expressed genes between CBMΦ and PBMΦ after polarization (FC > 4). **b, c** Enrichment analysis for the top enrichment GO terms from the biological process is shown for MΦ(IFN-γ) (**b**) and for MΦ(IL-10) (**c**), metabolic processes are highlighted in blue. **d** Schematic picture of metabolic enzymes that are downregulated in CBMΦ (in red) in glucose metabolism. **e–j** N-fold mRNA expression of metabolic enzymes in CBMΦ(IFN-γ) and CBMΦ(IL-10) compared with means of PBMΦ(IFN-γ) and PBMΦ(IL-10) analyzed by RT-qPCR. **e** *Glut1* mRNA expression was measured in $n = 3$ PBMΦ(IFN-γ), 3 CBMΦ(IFN-γ), 4 PBMΦ(IL-10), and 3 CBMΦ(IL-10) independent samples, **$p$-value = 0.0078. **f** *PFKM* mRNA expression was measured in $n = 3$ PBMΦ(IFN-γ), 3 CBMΦ(IFN-γ), 3 PBMΦ (IL-10), and 2 CBMΦ(IL-10) independent samples, *$p$-value = 0.0101. **g** *MPC1* mRNA expression was measured in $n = 7$ PBMΦ(IFN-γ), 4 CBMΦ(IFN-γ), 6 PBMΦ(IL-10), and 4 CBMΦ(IL-10) independent samples, *$p$-value = 0.0213. **h** *MPC2* mRNA expression was measured in $n = 3$ PBMΦ(IFN-γ), 3 CBMΦ (IFN-γ), 4 PBMΦ(IL-10), and 3 CBMΦ(IL-10) independent samples, **$p$-value = 0.0056. **i** *PC* mRNA expression was measured in $n = 3$ PBMΦ(IFN-γ), 3 CBMΦ(IFN-γ), 3 PBMΦ(IL-10), and 4 CBMΦ(IL-10) independent samples, *$p$-value = 0.0208 and **$p$-value = 0.0023. **j** *MDH2* mRNA expression was measured in $n = 3$ PBMΦ(IFN-γ), 3 CBMΦ(IFN-γ), 4 PBMΦ(IL-10), and 4 CBMΦ(IL-10) independent samples, **$p$-value = 0.0014 and ***$p$-value < 0.0001. For (**e–j**), bars indicate mean and error bars SD, two-tailed, one-sample test

compared with adult macrophages. We excluded any potential differences in background staining between adult and CB monocyte-derived macrophages by simultaneous staining with isotype controls (Supplementary Fig. 1F). In contrast, mTor mRNA expression was tendencially enhanced in cord blood-derived MΦ(IFN-γ) and cord blood MΦ(IL-10) macrophages (Supplementary Fig. 1G) compared with adult macrophages, which suggests that mTOR protein expression is reduced in cord blood-derived macrophages by the posttranscriptional mechanism. Furthermore, phosphorylation of the mTORC1 target S6 was decreased in MΦ(0) and MΦ(IL-10) and in MΦ(IFN-γ) cord blood-derived macrophages and the mTORC1 target 4E-BP1

phosphorylation at least in cord blood- derived MΦ(IL-10) macrophages (Fig. 4d, e). In conclusion, this suggests a generalized suppression of mTOR activation in cord blood-derived macrophages, which might account for the metabolic alterations.

**mTOR inhibition partially equalizes differences between PBMΦ and CBMΦ.** Rapamycin is an inhibitor of mTOR complex 1 (mTORC1), which enables us to analyze the effects of low mTOR activation in adult macrophages. In order to figure out if mTOR inhibition in adult macrophages shifts function and phenotype toward cord blood-derived macrophages, we performed

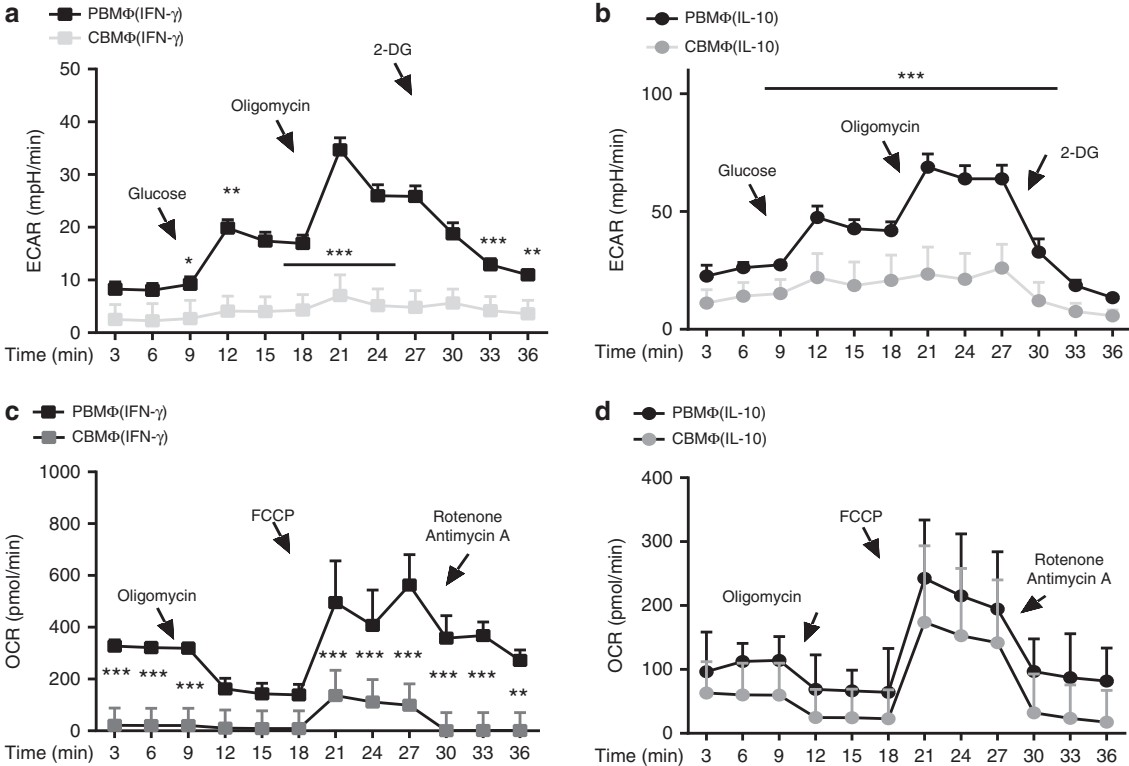

**Fig. 3** Cord-blood macrophages fail to induce glycolysis. ECAR measured under basal conditions and after addition of the indicated drugs in MΦ(IFN-γ) (**a**) and MΦ(IL-10) (**b**). Points indicate mean from three independent experiments (error bars represent standard deviation (SD)). OCR measured under basal conditions and after addition of the indicated drugs in MΦ(IFN-γ) (**c**) and MΦ(IL-10) (**d**). Points indicate mean from three independent experiments with SD ($N = 3$–4, heptaplicates $*p < 0.05$; $**p < 0.01$; $***p < 0.005$, one-way ANOVA, detailed statistics in supplemental informations)

whole-transcriptome analysis of rapamycin-treated adult MΦ (IFN-γ) and MΦ(IL-10) compared with untreated MΦ(IFN-γ) and MΦ(IL-10) macrophages. As expected, rapamycin altered the expression of several genes, which belong to metabolic processes in MΦ(IFN-γ) as well as MΦ(IL-10) adult macrophages (Fig. 5a, b). Interestingly, we also found a downregulation of genes that belong to the glycolysis pathway and the cholesterol pathway. Both pathways were downregulated in cord blood-derived MΦ(IL-10) macrophages as well (Fig. 5c, d). Nevertheless, gene expression analysis also uncovered several discrepancies between rapamycin-treated macrophages and cord blood-derived macrophages. KEGG pathway analysis of MΦ(IL-10) macrophages revealed that while Steroid biosynthesis, Fatty acid metabolism, and Pyruvate metabolism were enriched in both cord blood-derived MΦ(IL-10) and in rapamycin-treated macrophages compared with adult macrophages, other metabolic pathways found in CBMs were not significantly altered in rapamycin-treated macrophages. In detail, Carbon metabolism, Fatty acid degradation, Fatty acid elongation, Metabolic pathways, Tryptophan metabolism, and Propionate metabolism were not enriched after rapamycin treatment (Supplementary Table 1). In addition to metabolic pathways, CBMΦ also revealed differences compared with adult macrophages with regard to Notch signaling, Protein export, and Protein processing in the ER. This shows that beyond mTOR signaling, several other metabolic and non-metabolic pathways are differently regulated in cord blood macrophages as well. MΦ(IFN-γ) macrophages did not show any simultaneously regulated pathways compared with CBMΦ(IFN-γ) macrophages, which might be related to a high transcriptional variance within the CBMΦ(IFN-γ) macrophages (Supplementary Table 2). While rapamycin inhibited glycolysis significantly in MΦ(IFN-γ) (Fig. 5e), it only showed tendencies in MΦ(IL-10) (Fig. 5f). Thus, as mentioned above, other mTOR-

independent effects might be involved as well. To analyze whether mTOR inhibition also determines cytokine expression, we treated MΦ(IL-10) macrophages with the mTOR inhibitor rapamycin and incubated them with *E. coli*. Phagocytic indices did not differ in MΦ(IFN-γ) and MΦ(IL-10) from neonates and adults (Supplementary Fig. 3); however, MΦ(IL-10) from adults revealed a higher production of TNF-α, IL-6, IL-10, and IL-1β (Fig. 5g–j), which is in agreement with our observation that adult macrophages have a better capacity to become inflammatory than cord blood-derived macrophages and in agreement with the data from Ulas et al. that MyD88-dependent genes like TNF-α and IL-6 are expressed lower in newborn cord blood-derived monocytic cells treated with LPS[13]. MTOR inhibition by rapamycin furthermore downregulated TNF-α, IL-10, and IL-1β expression in adult MΦ (IL-10) (Fig. 5g–j). Rapamycin- treated adult MΦ(IFN-γ) also showed a reduction of TNF-α, IL-1β, and IL-6 secretion compared with untreated MΦ(IFN-γ), while IL-10 secretion was not different and very low in all MΦ(IFN-γ) (Fig. 5g–j). In contrast to adult macrophages, rapamycin treatment of cord blood macrophages did not result in any significant changes of cytokine expression. This further suggests that mTOR- mediated metabolic pathways are differentially regulated in adult and cord blood-derived macrophages.

**S100A8/A9 downregulates glycolysis and prevents polarization of PBMΦ.** To address the physiological mechanism behind this polarization defect, we first checked for the expression of S100A8/ A9 mRNA in adult and cord blood-derived macrophages. S100A8/A9 is highly abundant in neonatal serum and has broad transcriptional effects on neonatal monocytes[12,13]. Cord-blood-derived MΦ(IFN-γ) showed strongly enhanced S100A8/A9

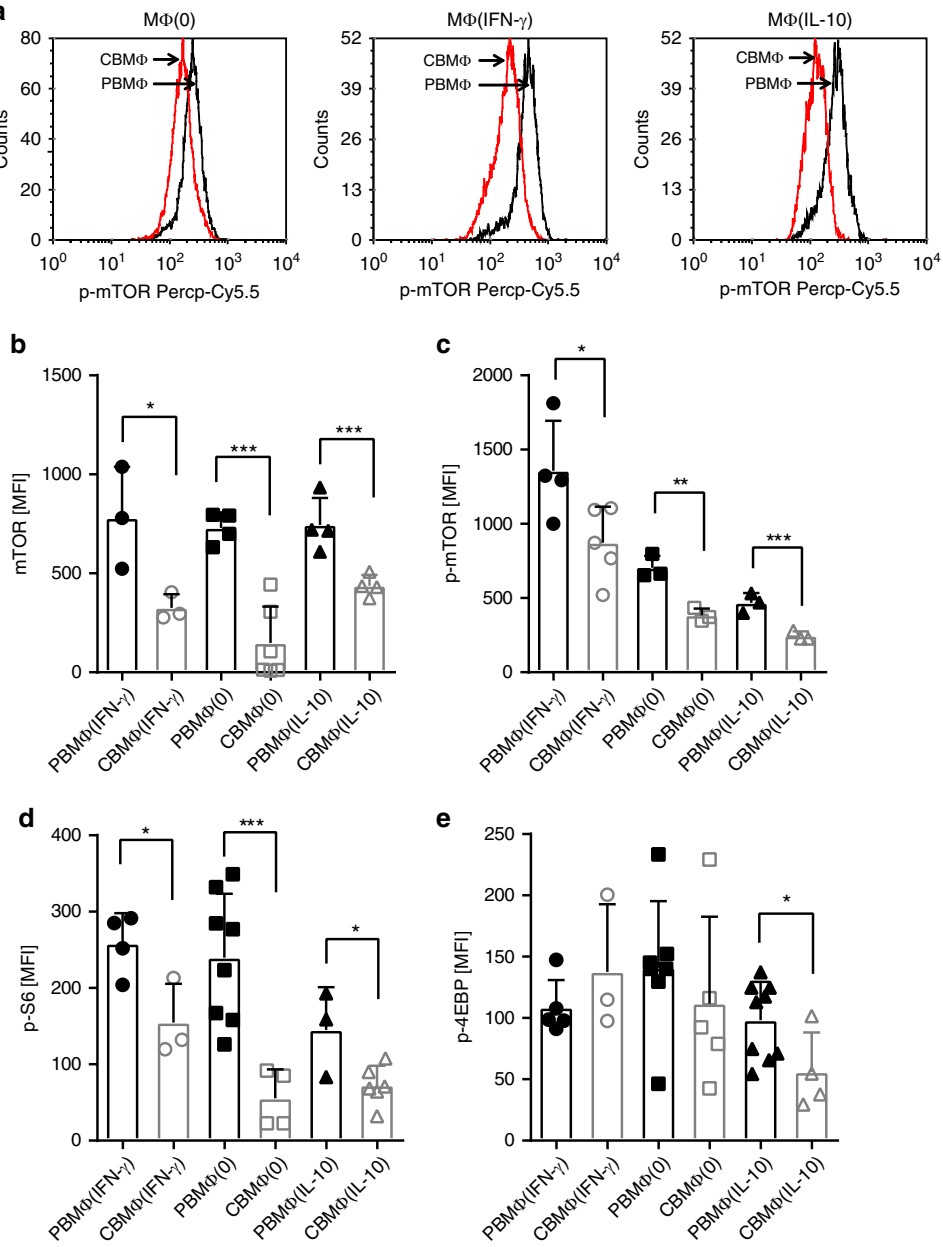

**Fig. 4** Cord-blood macrophages reveal a reduced activation of the mTOR pathway. Representative histograms showing (**a**) mean fluorescent intensity of p-mTOR expression in polarized CBMΦ (red) and PBMΦ (black). Statistics showing mean fluorescent intensity of total mTOR expression (**b**; $N = 3$-6), p-mTOR expression (**c**; $N = 3$-5), mTOR target genes pS6 (**d**), and p4-EBP (**e**; $N = 3$-10) in polarized CBMΦ (gray-framed bars) and PBMΦ (black-framed bars; *$p < 0.05$; **$p < 0.01$; ***$p < 0.005$), for statistics see supplemental information

mRNA levels compared with adult MΦ(IFN-γ), while the expression of S100A8/A9 in adult and cord blood MΦ(IL-10) was not different (Supplemental Fig. 2). Next, we incubated adult macrophages with M-CSF in the presence or absence of S100A8 or S100A9. Incubation of adult macrophages with S100A8 or S100A9 resulted in altered expression of macrophage surface markers, including HLA-DR, CD14, CD163, and CD206 and combined treatment with both S100A8 and S100A9 yielded similar results (Fig. 6b–e). Interestingly, the observed effects were comparable to rapamycin-mediated effects on surface expression (Fig. 6b–e). In addition and importantly, similar to incubation with rapamycin, incubation with S100A8/A9 solely or simultaneously resulted in a reduced expression of p-mTOR in MΦ(0) and MΦ(IFN-γ) (Fig. 6a), while no difference was noted in MΦ(IL-10), which might be related to the fact that IL-10 itself

downregulates p-mTOR expression. In line with this, the S100A8/A9 complex significantly and dramatically downregulated ECAR levels in adult MΦ(IFN-γ) and MΦ(IL-10) (Fig. 6f, g). This further implicates that enhanced S100A8/A9 expression in the serum of neonates might account for the phenotypic and metabolic alterations seen in cord blood macrophages. To further proof this, we incubated adult MΦ(IFN-γ) in the presence of cord blood serum (CS) with or without LPS-RS (Fig. 6h). LPS-RS is a penta-acetylated LPS derivate that blocks TLR4, a receptor of S100A8/9[19]. The reduced glycolysis caused by CS could be reversed by blocking the activity of S100A8 in CS via LPS-RS (Fig. 6h), while there was no effect on IL-10-treated macrophages (Supplemental Fig. 4). In conclusion, treatment of macrophages with S100 proteins mimics phenotypical and functional phenotypes and especially downregulates glycolysis of cord

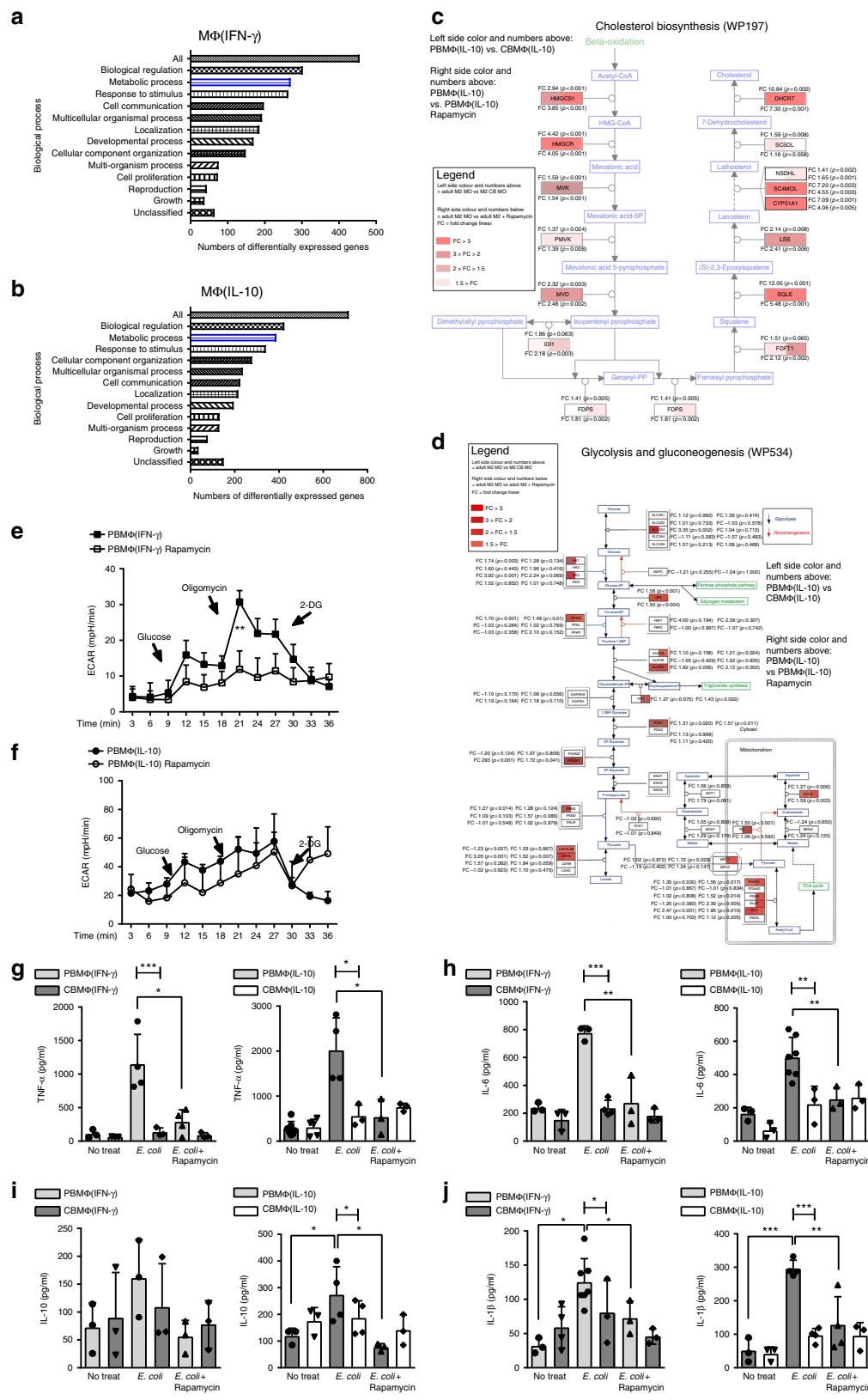

blood-derived macrophages and vice versa, but blocking S100 signaling abolishes CS-mediated glycolytic dysfunctions.

## Discussion

Macrophage activation is a key event in the inflammatory response and in septic states. Activated macrophages undergo profound reprogramming of their cellular metabolism. In this study, we show an altered activation of cord blood macrophages, irrespective of the stimulus used, paralleled by a reduced expression of cytokines and broad metabolic alterations, which in particular, affect glycolysis. This metabolic effect in cord blood macrophages can be partially mimicked by incubation of

**Fig. 5** mTOR inhibition in adult macrophages partly mimics phenotype and function of cord blood-derived macrophages. **a** Enrichment analysis for the top enrichment GO terms from the biological process is shown for rapamycin treated vs. untreated MΦ(IFN-γ) **b** and for rapamycin treated vs. untreated MΦ (IL-10) macrophages ($p < 0.05$, FC > 2). **c** Wikipathway analysis of differentially regulated genes within the cholesterol biosynthesis pathway. MΦ(IL-10) macrophages were compared with rapamycin-treated MΦ(IL-10) macrophages and CBMΦ(IL-10) macrophages. **d** Wikipathway analysis of differentially regulated genes within the glycolysis and gluconeogenesis pathway. MΦ(IL-10) was compared with rapamycin-treated MΦ(IL-10) macrophages and CBMΦ(IL-10). **e** ECAR measured under basal conditions and after addition of the indicated drugs in M1–MΦ. Points indicate mean from minimal three to four independent experiments (error bars represent SEM). **f** ECAR measured under basal conditions and after addition of the indicated drugs in M2–MΦ. Points indicate mean from three to four independent experiments (error bars represent SEM, **$p < 0.001$, one-way ANOVA). **g** MΦ(IFN-γ) and MΦ(IL-10) were incubated with *E. coli* +/− Rapamycin. TNF-α levels were measured by ELISA ($N = 3$–7; *$p < 0.05$, blunt-ended bars represent one-way ANOVA, ***$p < 0.001$). **h** MΦ(IFN-γ) and MΦ(IL-10) were incubated with *E. coli* +/− Rapamycin. IL-6 levels were measured by ELISA ($N = 3$–7; **$p < 0.01$, blunt-ended bars represent one-way ANOVA, **$p < 0.01$, ***$p < 0.001$). **i** MΦ(IFN-γ) and MΦ(IL-10) were incubated with *E. coli* +/− Rapamycin. IL-10 levels were measured by ELISA ($N = 3$–4; *$p < 0.05$, blunt-ended bars represent one-way ANOVA, *$p < 0.05$). **j** MΦ(IFN-γ) and MΦ(IL-10) were incubated with *E. coli* +/− Rapamycin. IL-1β levels were measured by ELISA ($N = 3$–7; **$p < 0.01$, ***$p < 0.001$, blunt-ended bars represent one-way ANOVA, **$p < 0.01$, ***$p < 0.001$; for detailed statistics see supplemental information)

adult macrophages with cord blood serum or incubation with S100A8/A9.

Several studies showed that metabolic shifts fuel multiple aspects of macrophage activation; moreover, that preventing these shifts impairs appropriate activation. Our study here suggests that broad alterations in energy metabolism are the reason for an altered polarization and phenotype of cord blood-derived macrophages. Especially, glycolysis is dramatically reduced in cord blood-derived macrophages. As the shift to glycolysis in response to LPS is important in the context of acute bacterial infection and utilization of glucose is associated with a proinflammatory macrophage phenotype[2], this might critically contribute to the altered polarization of cord blood-derived macrophages. Activation of the glycolysis pathway is accompanied by an activation of the mTOR pathway and we show here that this pathway is clearly diminished in cord blood macrophages, irrespective of the stimulus used. The mTOR pathway is critical for survival of fungal as well as *E. coli*-mediated sepsis[20], and our findings therefore could explain the heightened risk of sepsis and sepsis-mediated death in human neonates. Nevertheless, the most prominent pathogen in neonates is group B streptococci, which do not have LPS. We therefore chose IFN-γ to differentiate macrophages into a M1 phenotype. IFN-γ is a cytokine that is highly upregulated in neonates during sepsis. The same is true for IL-10 in the context of neonatal sepsis[15]; however, IL-10 expression is lower in neonatal CBMΦ than in PBMΦ (Fig. 5). Interestingly, human data on macrophage metabolism regarding these polarization procedures are scarce. There was one recent paper showing that IFN-γ induces a shift to glycolysis in human MΦ(IFN-γ). This shift to glycolysis is nevertheless much lower than in LPS-treated macrophages and therefore might explain the comparably low ECAR response in our setting (Fig. 3). With regard to IL-10, to our knowledge, this is the first study showing ECAR and OXPHOS levels in human macrophages solely treated with IL-10. There was a recent paper by Ip et al. showing an anti-inflammatory effect of IL-10 mediated by metabolic reprogramming of macrophages; however, in that setting, IL-10 was only used in combination with LPS[21] and it was shown that IL-10 signaling via STAT3 inhibits mTORC1 activation.

Such broad underlying alterations in neonatal metabolism must have a physiological reason. Neonates display extremely high levels of S100A8/A9 proteins[12]. S100A8/A9 are proteins secreted by neutrophils that are characterized by a high bactericidal activity and therefore act as a barrier just in front of the first cellular contact. S100A8/A9 itself is an endogenous TLR4 agonist and able to activate human macrophages. In patients with systemic onset juvenile idiopathic arthritis, S100A8/A9 levels are also highly elevated and these patients undergo severe systemic inflammation, which is not the case in neonates[22]. It might therefore be that the glycolysis pathway is shut down in neonatal macrophages to

prevent a hyperinflammatory state induced by pathogens and also commensal bacteria that colonize the neonate in the first few days after birth. The high abundance of S100A8/A9 might be a critical mechanism which reduces glucose metabolism in neonatal macrophages. It was shown before that loss of S100A8/A9 in mice resulted in severe septic hyperinflammation in a model of neonatal Staph. aureus infection with a strongly enhanced mortality[13]. In addition, low expression of S100A8/A9 in neonatal serum was associated with enhanced susceptibility toward sepsis. In our hands, incubation of macrophages with S100A8/A9 results in shutdown of glycolysis and downregulation of p-mTOR in PBMΦ (0) and MΦ(IFN-γ). In MΦ(IL-10), S100A8/A9 did not further reduce p-mTOR expression; however, IL-10 itself suppresses p-mTOR expression[21] and the effect of S100A8/A9 on the expression of MΦ(IL-10) markers might be independent of p-mTor. In line with this, mRNA expression of S100A8/A9 was strikingly enhanced only in MΦ(IFN-γ) cord blood macrophages. In addition to this, mTOR inhibition with rapamycin only partially mimicked S100 effects on adult PB macrophages. While glycolysis was reduced significantly in MΦ(IFN-γ) and tendencially in MΦ (IL-10), we did not observe the effects on OXPHOS. Gene expression analysis also showed similarities between cord blood-derived macrophages and rapamycin-treated adult macrophages, especially with regard to metabolic pathways, such as glycolysis and cholesterol biosynthesis, but also several differently regulated genes and pathways. This further implicates that S100A8/A9 effects are in part independent of p-mTOR. The delineation of these additional pathways will be a topic for further investigations. LPS itself also induces p-mTOR and mTOR-mediated pathways include the activation of glycolysis, when signaling via TLR4. However, in states of immunoparalysis due to sepsis or during LPS-induced tolerance, downregulation of glycolysis occurs in human monocytic cells[20]. Thus, our data suggest that S100A8/A9 acts in a way similar to the mechanism of LPS tolerance and suppresses glycolysis. On the one hand, this prevents hyperinflammation in neonatal sepsis; on the other hand, newborns might be more prone to septicemia per se, since glycolysis as the rapid response tool toward bacterial infection is altered and neonates are therefore in a certain state of immunoparalysis, which might also be related to the pathogen. Newborns come from a sterile intrauterine surrounding before birth and rapidly undergo colonization of the mucosal surfaces and the skin within the first 48–72 h. During this period, the host has to adapt to this bacterial load without induction of inflammation. This physiological adaptive measure, downregulation of glycolysis, also might keep macrophages in a tolerized state and prevents fatal hyperinflammation shortly after birth due to the immediate bacterial colonization of the newborn[13].

It will also be of major interest, at which time point during development, macrophage polarization in infants is changed

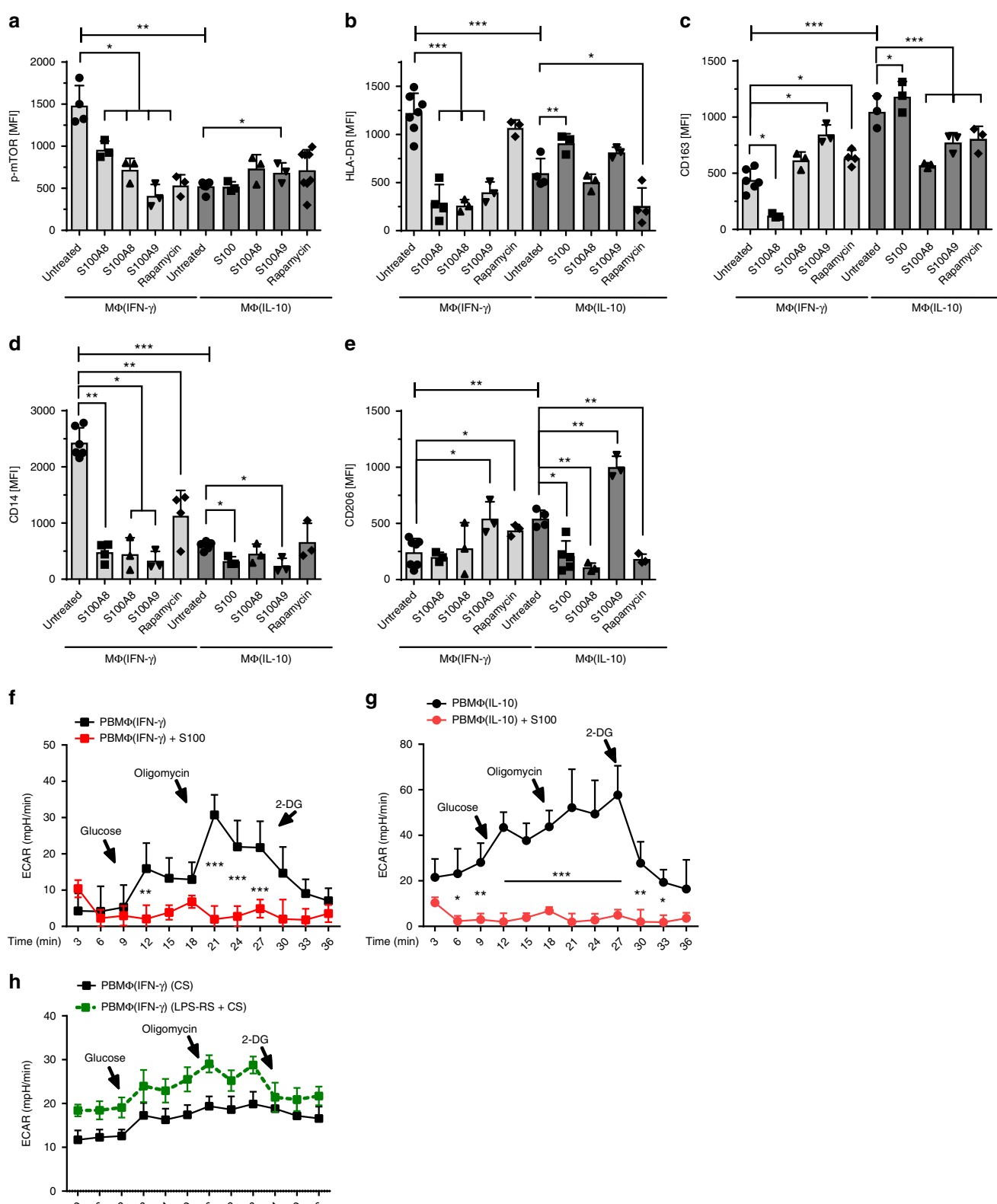

toward an adult phenotype. Usually, neonates are at risk of developing late-onset sepsis until 6 weeks after birth. It is tempting to speculate that this is the time point at which macrophage polarization is altered toward a more adult phenotype. However, this will be a challenge to prove, since it will be an ethical problem to get blood from healthy babies of that age group.

The issue that we cannot directly analyze macrophages from newborns or a few-day-old babies is also the main limitation of this study. We therefore have to interpret our findings, which result from ex vivo incubation and activation in a different milieu than what would be expected in the newborn with caution. Further studies with macrophages from newborn mice might be helpful to verify our hypothesis.

**Fig. 6** S100A8/A9 downregulates p-mTOR, activation markers, and glycolysis in PBMΦ. Bar diagrams show expression of pmTOR (**a**), expression of MΦ(IFN-γ) surface proteins (HLA-DR and CD14) (**b**, **d**) and MΦ(IL-10) surface proteins CD206 and CD163 (**c**, **e**) in untreated macrophages and macrophages treated either with S100A8 or S100A9 or treated with both simultaneously or treated with rapamycin. Polarization was induced in PBMC using 100 ng/ml hM-CSF for 72 h in RPMI in the presence or absence of 5 μg/ml S100A8 or A9. Further polarization was achieved by addition of 50 ng/ml IFN-γ MΦ(IFN-γ) or 10 ng of IL-10 MΦ(IL-10) for an additional 48 h. Immunotyping was performed by flow cytometric analysis detecting selected surface markers of mTOR, CD14, HLA-DR, CD163, and CD206. Significance of reduced expression in S100A8 or S100A9-treated MΦ was tested by utilizing the ANOVA test (indicated by blunt-ended bars) followed by Bonferroni post test ($N = 3$–8; *$p < 0.05$; **$p < 0.01$, ***$p < 0.005$). ECAR levels were measured after incubation of MΦ(IFN-γ) (**f**) and MΦ(IL-10)Φ with S100A8/A9 (**g**). ECAR levels were measured after incubation of MΦ(IFN-γ) with CB serum (CS) +/− LPS-RS. Points indicate mean from three to six experiments, error bars SEM (**h**). For detailed statistics see supplemental information

As a conclusion, neonatal macrophages display altered polarization capacities and broad metabolic defects, which are related to reduced mTOR activation and enhanced expression of S100A8/A9, that induce a metabolic state of LPS tolerance. These findings might result in a higher risk of neonatal sepsis as well as a risk of metabolic alterations during sepsis and might offer promising targets for therapeutic interventions.

## Methods

**Patients**. The study protocol was approved by the Ethics Committees of Aachen University Hospital (Permission No: EK150/09, Oct. 6, 2009, signed by Profs. G. Schmalzing and U. Buell, respectively). All adult participants involved gave written consent to use their blood samples. All term neonates (in total 26 unique cord blood samples) were delivered spontaneously and did not exhibit signs of infection, as defined by clinical status, white blood cell count, and C-reactive protein. Mothers with amnion infection and prolonged labor (>12 h) were excluded. Umbilical cord blood was placed in heparin-coated tubes (4 IE/ml blood), immediately following cord ligation. In brief, cord blood was diluted with three volumes of PBS and loaded in a density gradient (Ficoll Histopaque) and centrifuged at 300 x g for 15 min at RT (no brake). The leukocyte-enriched interphase was washed twice with PBS and was resuspended in RPMI supplemented with 10% fetal calf serum.

**Antibodies and reagents**. Antibodies to CD14 (clone MEM18, Immunotools, 21270146, 1:500), CD16 (clone B73.1, BD Biosciences, 332779, 1:200), CD32 (clone FLI8.26, BD Biosciences, 552884, 1:200), CD64 (clone FCGR1, BD Biosciences, 555527, 1:500), CD163 (clone GH1/61, BD Biosciences, 563887, 1:250), HLA-DR (clone G46–6, BD Biosciences, 560896, 1:100), CD206 (clone 19.2, BD Biosciences, 550889, 1:100), TLR4 (clone HTA-25, BD Biosciences, 555657, 1:200), Arginase-1 (clone sL6Arg, Thermo Fisher Scientific, 1:100), phospho-STAT3 (clone LUVNKLA, BD Biosciences, 612569, 1:200), phospho-mTOR (clone MRRBY, Thermo Fisher Scientific, 12-9718-42, 1:200), phospho-S6 (clone cupK43K, Thermo Fisher Scientific, 17-9007-42, 1:200), phospho-4EBP (clone V3NTY24, Thermo Fisher Scientific, 50-9107-42, 1:200), and Ig-matched controls were from BD Biosciences (Heidelberg, Germany), Thermo Fisher Scientific (formerly eBioscience; Waltham, MA, USA), and Immunotools (Friesoythe, Germany), respectively. Antibodies were diluted according to the manufacturer's recommendations. Cytokines IFN-γ, IL-10, and M-CSF were from PAN Biotech (Aidenbach, Germany). Rapamycin was purchased from Thermo Fisher Scientific and added at a concentration of 50 nM for 1 h before harvesting. S100A8 and S100A9 homodimers were recombinantly expressed in *E. coli* BL21 (DE3) cells[23]. Harvested bacteria were lysed and the prepared inclusion bodies were dissolved in 8 M urea buffer. For correct refolding, samples were adjusted to pH 2.0, followed by stepwise dialysis to pH 7.4 in the presence of 2 mM DTT. Samples were applied to an anion exchange column and gel filtration chromatography. Possible endotoxin contaminations were determined by a Limulus amebocyte lysate assay (BioWhitaker, Walkersville, MD) and could either not be detected or were below 1 pg of LPS/μg homodimer in the different batches.

**Differentiation protocols**. Our polarization protocol followed the protocols, published earlier[24]. In brief, leukocytes were prepared by density-gradient centrifugation of whole blood from healthy adult donors and cord blood from term[25]. Polarization was induced by seeding $5 \times 10^5$ cells/ml in 12-well tissue culture cells and administration of 100 ng/ml M-CSF for 72 h in RPMI (designated as MΦ(0)). Where indicated, further polarization was achieved by addition of 50 ng/ml IFN-γ (designated as MΦ(IFN-γ), 10 ng of IL-10 (designated as MΦ(IL-10) for an additional 48 h. S100A8 or S100A9 was given in addition to M-CSF for 50 h in a final concentration of 5 μg/ml each. In indicated experiments, serum from 1-ml full cord blood as well as adult peripheral blood (designated as CS and AS, prepared by centrifugation at $200 \times g$ for 15 min) and CS supplemented with LPS-RS (50 ng/ml) was added to a final concentration of 20% (v/v) for an additional 3 days.

**Phagocytosis**. *E. coli* DH5α, an encapsulated K12 laboratory strain, carrying the green fluorescent protein (*gfp*)-mut2 gene (*E. coli*-GFP) was a generous gift from Prof. Dr. Dehio (University of Basel, Switzerland) and was used for phagocytosis as previously described. Bacteria were freshly grown in Lennox-L-Broth medium (Invitrogen) until early logarithmic growth, resuspended in PBS, and used immediately. Infection was performed at a multiplicity of infection (MOI) of 25:1, which was achieved by dilution with PBS. The phagocytosis assays were performed by co-incubating the appropriate MΦ with a freshly grown *E. coli*-GFP culture resuspended in PBS. Dilution was always a plaque-forming unit (pfu) of $2.5 \times 10^6$ *E. coli* to $10^5$ MΦ (MOI (multiplicity of infection) 25) for 1 h at standard cultivation conditions. Afterward, the medium was replaced by fresh medium. The phagocytosis index (CD14+GFP+ MΦ: CD14+ MΦ) was analyzed by flow cytometry.

**Detection of surface receptors and intracellular signal transducers**. Macrophages were detached from culture dishes by incubation in PBS/EDTA (10 mM final conc.) for 10 min at 37 °C. For staining of surface molecules, cells were fixed in 1% paraformaldehyde for 1 h. Afterward, the cells were washed and blocked with PBS/FCS (5% v/v) for 20 min at RT and were stained with fluorochrome-labeled antibodies in PBS/FCS (5% v/v) that took place for 60 min at RT followed by additional washing. Intracellular staining of blood-derived macrophages was started by using the Foxp3 transcription factor staining kit from Thermo Fisher Scientific following the manufacturer's recommendations.

**Flow cytometry**. A daily calibrated FACS-Canto flow cytometer (Becton Dickinson, MountainView, CA) was used to perform phenotypic analysis. Macrophages were gated by forward (FSC) and side scatter (SSC) expression, as shown in Supplementary Fig. 1. For data analysis, FCS-Express 4.0 Research Edition (DeNovo software Glendale, CA, USA) was used.

**Transcriptome analysis**. To further identify pathways that are responsible for phenotypic differences between PBMΦ and CBMΦ, we performed an Affymetrix-based transcriptome analysis (HTA2, arraysGeneChip® Human Transcriptome Array 2.0, Affymetrix, Santa Clara, CA, USA). Therefore, $3 \times 10^6$ macrophages were lyzed and subjected to RNA purification according to the manufacturer's recommendations. The Qiagen RNA preparation kit was used (Qiagen, Erkrath, Germany). The quantity of total RNA was measured using a NanoDrop ND-1000 Spectrophotometer (Thermo Scientific, Waltham, MA, USA). Optical density ratios at 260/280 were consistently above a certain threshold. The total RNA quality was assayed with a RNA 6000 Nano Kit on an Agilent BioAnalyzer (Agilent Technologies, Santa Clara, CA, USA). Only samples with intact, distinct ribosomal peaks were taken. Microarray data were analyzed using MicroArray Suite Software 5.0 (Affymetrix) and further studied analyzing the Expressionist Suite software package (Gene Data), which allows identification of genes that are significantly regulated in multiple independent experiments. Transcription rates were considered to be significantly regulated once a *p*-value of lower than 0.05 and a twofold difference was observed.

**ELISA**. The TNF-α and IL-10 enzyme-linked immunosorbent assays (ELISA) were purchased from Ebiosciences (Ebiosciences-Natutec, Frankfurt, Germany). The IL-1-β and IL-6 enzyme-linked immunosorbent assays (ELISA) were purchased from Immunotools (Immunotools, Friesoythe, Germany). All ELISA kits were used according to the manufacturer's recommendations. The readout was executed in a spectra max 340PC ELISA reader (Molecular Devices, Sunnyvale, CA, USA) with a sensitivity from 4 to 500 pg/ml.

**RNA isolation and real-time PCR**. Total RNA was isolated using the RNeasy Mini Kit (Qiagen, Germany). cDNA was then generated from 200 ng of total RNA using the RevertAid H Minus First Strand cDNA Synthesis Kit (Thermo Fisher Scientific, USA) according to the manufacturer's instructions. qRT-PCR was performed using the SYBR Green PCR kit (Eurogentec, Germany) and data were acquired with the ABI prism 7300 RT-PCR system (Applied Biosystems/Life Technologies, Germany). Each measurement was set up in duplicate. After normalization to the endogenous reference control gene β-actin, the relative expression was calculated. The following primers were used for the genes indicated: *β-Actin*, forward primer: GACTACCTCATGAAGATCCTCACC; reverse primer: TCTCCTTAA TGTCACGCACGATT; *Glut1*, forward primer: CTTTGGCCGGCGGAATTCAA;

reverse primer: CCCAGTTTCGAGAAGCCCAT; *MDH2*, forward primer: AGCA
CCGGAAGAGTCGCT; reverse primer: CTTCCCCAGCTGTTCTCTGAGG;
*MPC1*, forward primer: ACTATGTCCGAAGCAAGGATTTC; reverse primer:
CGCCCACTGATAATCTCTGGAG; *MPC2*, forward primer: TACCACCGGC
TCCTCGATAAA; reverse primer: TATCAGCCAATCCAGCACACA; *PC*,
forward primer: GCTGGAGGAGAATTACACCCG; reverse primer GGATGTTC
CCATACTGGTCCC; *PFKM*, forward primer: AGCACCGGAAGAGTCGCT;
reverse primer: CTTCCCCAGCTGTTCTCTGAGG; *S100A8*, forward primer: TGC
TAG AGA CCG AGT GTC CTC AG; reverse primer: CCA TCT TTA TCA CCA
GAA TGA GGA AC; *S100A9*, forward primer: TTC AAA GAG CTG GTG CGA
AAA G; reverse primer: GCA TTT GTG TCC AGG TCC TCC.

**Seahorse assay**. In total, $2 \times 10^5$ cells were seeded on gelatin-coated plates and
OCR/ECAR were measured using the XF96 Extracellular Flux Analyzer (Seahorse
Bioscience) following the manufacturer's instructions. OCR was measured in XF
media containing 11 mmol/L glucose and 1 mmol/L sodium pyruvate under basal
conditions and in response to 1 μmol/L oligomycin, 1 μmol/L carbonyl cyanide p-
trifluoromethoxyphenylhydrazone (FCCP), and 0.1 μmol/L rotenone plus 0.1
μmol/L antimycin A. Extracellular acidification rate (ECAR) was measured in assay
medium (DMEM supplemented with 4.5 g/l glucose and 2 mM glutamine) under
basal conditions and in response to 10 mM glucose, 1 M oligomycin, and 100 mM
2-deoxyglucose.

**Statistical analysis**. The results are expressed as mean +/− standard deviation.
Error bars represent standard deviations unless indicated as SEM. Values of $p <$
0.05 were considered significant. Analyses were done with statistical software
performing unpaired (Mann–Whitney U test) Student's *t* test and two-way
ANOVA adjusted according to Bonferroni–Holm for multiple group comparisons,
as provided by GraphPad Software Statistical Package, La Jolla, CA 92037, USA.

**Reporting summary**. Further information on experimental design is available in
the Nature Research Reporting Summary linked to this article.

## Data availability
The datasets that support the findings of this study are available in GEO Database. They
have been assigned GEO accession numbers GSE 124105 and GSE 124020. Further data
that support the findings of this study are available within the article and Supplementary
Files. All other data, including raw data used in each figure will be provided upon
reasonable request to the corresponding author.

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

## Author contributions
All authors were involved in drafting the article or revising it critically for important
intellectual content, and all authors approved the final version to be published. S.D., K.O.,
J.M., M.L., B.D. and I.C. contributed to the acquisition of data, analysis, and inter-
pretation of data. J.R., T.V. and D.V. contributed to the study design, provided S100A8
and A9 protein, and provided critical feedback on intellectual content. K.T. and T.O.
conceived the study and wrote the paper. S.D. and K.O. contributed equally. K.T. and
T.O. contributed equally.

## Additional information
019-09359-8.

**Competing interests:** The authors declare no competing interests.

**Reprints and permission** information is available online at http://npg.nature.com/
reprintsandpermissions/

**Journal peer review information:** *Nature Communications* thanks the anonymous
reviewers for their contribution to the peer review of this work. Peer reviewer reports are
available.

**Publisher's note:** Springer Nature remains neutral with regard to jurisdictional claims in
published maps and institutional affiliations.

