## [Peer Review File · Nature Communications]

Reviewers' comments:

Reviewer #2 (Remarks to the Author):

Dreschers et al compare the metabolic and gene expression profile of polarized macrophages derived from either adult or cord blood monocytes. Several differences can be noticed, possibly due to a reduced activation of the mTOR pathway in neonatal macrophages. Several main concerns remain:

- 1) The authors still refer to older nomenclature (eg M2a, b, c), while there are more recent efforts to classify macrophage activation states (refer to Murray et al, Immunity, 2014)
- 2) The authors use terms such as "adult undifferentiated macrophages", "neonatal macrophages"...It should be made clear that all studies were performed on blood monocyte-derived macrophages, generated in vitro in the presence of CSF1.
- 3) The use of MFI as read-out for protein expression (determined via FACS) is problematic. First of all, it is not specified whether the authors use Mean or Median Fluorescence Intensity. Median is preferred. In addition, no isotype controls seem to be used, hence any potential differences in background staining between adult and CB monocyte-derived macrophages remains obscured. The authors should use a delta-MFI as read-out (= MFI of marker – MFI of isotype control).
- 4) Fig 2. It would be convenient to indicate which genes are implicated in which metabolic pathway.
- 5) There is no reference to Figures 3 and 4 in the text.
- 6) mTOR activation: This text is too minimal to describe the results. Also P-mTOR staining lacks isotype controls. Moreover, no information is given about total mTOR levels, which may already be different at baseline between adult and CB macrophages.
- 7) mTOR inhibition: the effect on only two cytokines is shown. This is too minimal. A profound gene expression analysis should be performed in mTOR inhibited adult macrophages. Effects on the metabolic profile should be determined. Moreover, how can mTOR inhibition upregulate TNF in macrophages which are already very low in mTOR activation? One would only expect an effect at the level of mTOR(hi) adult Mph.
- 8) Fig 6: This is very indirect evidence and does not prove an effect of S100A8/9 in vivo. Better would be to assess the effect of neonate bodily fluid (cord blood serum?) on macrophages with or without S100A9/8 blockade
- 9) Quite heterogeneous effects of S100A8 versus S100A9 can be seen. For example on CD206 expression. S100A8/A9 also, and perhaps mainly, functions as a heterodimer. The authors should test this.

Reviewer #3 (Remarks to the Author):

In this work, Dr. Drescher and Colleagues have performed whole transcript profiling and metabolic analyses on polarized macrophages derived from cord blood mononuclear cells taken from term neonates born by spontaneous vaginal delivery. It is recognized why it was used, but there are significant limitations to results derived from cord blood only.

1. Distracting statements of "fact" are made throughout the manuscript introduction, results, and discussion without appropriate supporting references. If there are "number of references" restrictions for the journal, perhaps a complete list can be offered in the online version.
2. Neonatal cord blood does not reflect a state of health and it is difficult if not impossible to gauge the generalizability of the findings of a study focused on cord blood. This tissue reflects a transient state that one would expect with the stress of delivery, yet a state that is more persistent throughout the first days/weeks of life is implied/inferred. As a corollary, would the results on blood drawn from the mother immediately after she gave birth also be interpreted as broadly

reflective of women? The authors should comment on the limitation of using this tissue. The practice of collecting and studying cord blood largely results from the blood volume restrictions on newborns after birth, particularly in the extremely low birth weight infant (<1 kg), which may have total circulating blood volume of <50mL. Cord blood is used as a "low hanging fruit" rather than because it is informative.

3. The data suggests that there is an inherent defect in broad deficits in energy metabolism, and especially glycolysis, that affects polarization and function of neonatal macrophages. The challenges of isolating macrophages from humans especially neonates are clear. However, the artificiality of incubating CBMCs/PBMCs for 3 days after harvest and gradient separation, and cytokine polarization, then examination certainly represents a different milieu and stimulus than what would be expected in the newborn. Can the authors comment on the limitations of findings achieved in this way? If the findings are to be generalized to newborn immune function, it is worth having the authors comment on how their findings fit into the fact that not all newborns get infected, if fact most do not. Did they find differences among the neonatal patients ("n" was not given-see below) they examined?

4. It is unclear how many newborns were included in the analyses. E.g. if this data was derived from <10 infants, do the authors suggest the findings could be generalized? Other demographic variables would also be informative (sex, race, etc).

5. Several figures are incomplete or have abbreviations that are not spelled-out in the legend. Fig 3 is an example. Fig 4. Where are letters B/C?

7. Phagocytic indices are described in the results as being measured but they are not described in methods. E. coli incubation (ATCC strain?, CFUs?, duration, media?) also was described in results, and not in methods.

8. The authors suggest that S100A8/A9 expression is the mechanism behind the mTOR-dependent findings but did not measure S100 proteins nor defend why S100 proteins would still be over expressed 4 days after the cells were removed from the cord blood. Were those genes being over-expressed in the transcript?

Minor

1. "Third world" is inappropriate. It would be better to refer to these areas as low-middle income countries (LMIC) or "under resourced".

2. There is a sentence "...glycolysis, although less efficient in generating ATP, ..." that is repeated 3 times nearly verbatim in the paper.

Gavin Mason, PhD
Associate Editor
Nature Communications

Klinikdirektor
Univ.-Prof. Dr. med. Norbert Wagner

Universitätsklinikum Aachen
Anstalt öffentlichen Rechts (AÖR)
Pauwelsstraße 30
52074 Aachen
www.ukaachen.de

Sekretariat
Tel.: 0241-80-88700
Tel.: 0241-80-88701
Fax: 0241-80-82492
kinderklinik@ukaachen.de
www.kinderklinik.ukaachen.de

Aachen, den 26.10.2018

In particular we would like to thank the reviewers for their critical ideas that helped to improve our manuscript. We performed a number of additional experiments to address their comments and thoroughly revised the manuscript. A point by point reply follows:

Reviewer #2 (Remarks to the Author):

Dreschers et al compare the metabolic and gene expression profile of polarized macrophages derived from either adult or cord blood monocytes. Several differences can be noticed, possibly due to a reduced activation of the mTOR pathway in neonatal macrophages. Several main concerns remain:

1) The authors still refer to older nomenclature (eg M2a, b, c), while there are more recent efforts to classify macrophage activation states (refer to Murray et al, Immunity, 2014)

We changed this accordingly in the revised manuscript. The term “activated” instead of polarized is used and macrophages are now classified as M(IFN- γ) and M(IL-10), which is consistent with the nomenclature linked to activation standards as suggested by Murray et al., 2014 (lines 84-91).

2) The authors use terms such as “adult undifferentiated macrophages”, “neonatal macrophages”...It should be made clear that all studies were performed on blood monocyte-derived macrophages, generated in vitro in the presence of CSF1.

We clarified this point in the revised manuscript.

3) The use of MFI as read-out for protein expression (determined via FACS) is problematic. First of all, it is not specified whether the authors use Mean or Median Fluorescence Intensity.

Vorsitzender des Aufsichtsrates
Dr. Robert G. Gossink

Vorstandsvorsitzender
Prof. Dr. med. Thomas H. Ittel

Kaufmännischer Direktor
Dipl.-Kfm. Peter Asché

Sparkasse Aachen · BIC: AACSD33
BLZ: 390 500 00 · Kto.: 13 004 015
IBAN: DE27 3905 0000 0013 0040 15
Commerzbank AG · BIC: DRESDEFF390
BLZ: 390 800 05 · Kto.: 203 309 400
IBAN: DE79 3908 0005 0203 3094 00
UST-IdNr.: DE813100566

Median is preferred. In addition, no isotype controls seem to be used, hence any potential differences in background staining between adult and CB monocyte-derived macrophages remains obscured. The authors should use a delta-MFI as read-out (= MFI of marker - MFI of isotype control).

The flow cytometric data in Figure 1 already represent G-Mean delta-MFI (= MFI of marker - MFI of isotype control) data. We used G-Mean to determine expression. This information was added in the Figure Legend. The alternative use of median did not alter the results. The gating strategy was added (Supplemental Figure 1).

4) Fig 2. It would be convenient to indicate which genes are implicated in which metabolic pathway.

We added a schematic picture in Figure 2 which shows transcriptionally altered genes and their function during glycolysis. In addition to this we now describe functions of altered genes within the text.

5) There is no reference to Figures 3 and 4 in the text.

We changed this accordingly.

6) mTOR activation: This text is too minimal to describe the results. Also P-mTOR staining lacks isotype controls. Moreover, no information is given about total mTOR levels, which may already be different at baseline between adult and CB macrophages.

We added a more comprehensive description of our results; we also added total mTOR levels (Fig. 4B), which already show differences and we discussed it in the text (lines 146-156). In addition we now show isotype staining in Supp. Fig. 1

7) mTOR inhibition: the effect on only two cytokines is shown. This is too minimal.

IL6 und IL1 β ELISA data were added (Fig. 5 G and 5 J), which, in addition to the TNF and IL-10 data show significant differences between M(IFN- γ) and M(IL-10) macrophages from adults and cord blood derived macrophages after E.coli activation. The exception is the IL-10 secretion in M(IFN- γ) macrophages which showed no significant increase after E.coli activation in macrophages of newborn and adults. Rapamycin treatment downregulated the cytokine expression in adult macrophages with the exemption of IL-6. Rapamycin did show no further effects/downregulation in macrophages polarized from cord blood monocytes. It is possible that in macrophages derived from cord blood monocytes other mTOR independent pathways are

more active than in the adult counterparts. Therefore we extended our studies as pointed out next.

A profound gene expression analysis should be performed in mTOR inhibited adult macrophages. Effects on the metabolic profile should be determined. Moreover, how can mTOR inhibition upregulate TNF in macrophages which are already very low in mTOR activation? One would only expect an effect at the level of mTOR(hi) adult Mph.

We performed a substantial analysis of mTOR inhibition in adult macrophages. To this end, we analyzed cytokine secretion, metabolic profiles and a whole transcriptome gene expression analysis. In detail, in the revised manuscript we provide cytokine data in M(IL-10) and also in M(IFN- γ) and demonstrate that Rapamycin regulates IL-10, IL-6, TNF- α and IL-1b production (Fig. 5G-J). ECAR levels were downregulated by Rapamycin in M(IFN- γ) macrophages (Fig. 5E) but only tendentially in M(IL-10) macrophages (Fig. 5F). Microarray analysis also uncovered similarities between Rapamycin treated adult peripheral blood derived macrophages and CB derived macrophages (Fig. 5A - D), however several genes and pathway were differentially regulated. Adding this to the fact that S100A8/A9 showed a more convincing effect on ECAR and OCR levels (Fig. 6) we suppose that S100A8/A9 mediated effects are not fully mTOR dependent, but might include other signaling pathways as well that determine CB macrophage phenotype and function.

8) Fig 6: This is very indirect evidence and does not prove an effect of S100A8/9 in vivo. Better would be to assess the effect of neonate bodily fluid (cord blood serum?) on macrophages with or without S100A9/8 blockade

We performed seahorse assays and incubated adult macrophages with cordblood serum, which resulted in decreased glycolysis (Fig.6H). A partial resolution of this phenotype was achieved using a blocking reagent (TLR4 antagonist LPS-RS) (Fig. 6H). This was only significant in IFN- γ induced macrophages and there were tendencies in IL-10 induced macrophages. However direct incubation with S100A8/A9 heterodimers produced a strong phenotype (Figure 6F and G, see also question 9)

9) Quite heterogeneous effects of S100A8 versus S100A9 can be seen. For example on CD206 expression. S100A8/A9 also, and perhaps mainly, functions as a heterodimer. The authors should test this.

In addition to the flow cytometric analysis, we performed seahorse assays of adult macrophages generated in the presence of S100A8/A9 heterodimers which convincingly show a phenotype in ECAR that closely resembles the phenotype of cord blood, suggesting that S100A8/A9 is indeed

one major component of the phenotype that we find in cord blood derived macrophages (Fig. 6F and G).

Reviewer #3 (Remarks to the Author):

In this work, Dr. Drescher and Colleagues have performed whole transcript profiling and metabolic analyses on polarized macrophages derived from cord blood mononuclear cells taken from term neonates born by spontaneous vaginal delivery. It is recognized why it was used, but there are significant limitations to results derived from cord blood only.

1. Distracting statements of "fact" are made throughout the manuscript introduction, results, and discussion without appropriate supporting references. If there are "number of references" restrictions for the journal, perhaps a complete list can be offered in the online version.

The manuscript was intensely revised and references were added.

2. Neonatal cord blood does not reflect a state of health and it is difficult if not impossible to gauge the generalizability of the findings of a study focused on cord blood. This tissue reflects a transient state that one would expect with the stress of delivery, yet a state that is more persistent throughout the first days/weeks of life is implied/inferred. As a corollary, would the results on blood drawn from the mother immediately after she gave birth also be interpreted as broadly reflective of women? The authors should comment on the limitation of using this tissue. The practice of collecting and studying cord blood largely results from the blood volume restrictions on newborns after birth, particularly in the extremely low birth weight infant (<1 kg), which may have total circulating blood volume of <50mL. Cord blood is used as a "low hanging fruit" rather than because it is informative.

We discuss the limitation of using cord blood derived macrophages instead of newborn peripheral blood derived macrophages in the discussion of our revised manuscript. Indeed cord blood is a "low hanging fruit" but the experimental alternatives described in the literature are rare. Small volumes of peripheral blood taken by venipuncture from newborns were indeed described for comparative studies with monocytes. Newborn blood samples would be much closer to the in-vivo situation but would be more stressing for the small patients. Moreover, it is questionable whether this more native source of macrophages is worth its price for studies which are primarily performed in-vitro. Nevertheless, we described the limitations and alternative experimental procedures which take account of your critics in the discussion section.

3. The data suggests that there is an inherent defect in broad deficits in energy metabolism, and especially glycolysis, that affects polarization and function of neonatal macrophages. The challenges of isolating macrophages from humans especially neonates are clear. However, the artificiality of incubating CBMCs/PBMCs for 3 days after harvest and gradient separation, and

cytokine polarization, then examination certainly represents a different milieu and stimulus than what would be expected in the newborn. Can the authors comment on the limitations of findings achieved in this way? If the findings are to be generalized to newborn immune function, it is worth having the authors comment on how their findings fit into the fact that not all newborns get infected, if fact most do not. Did they find differences among the neonatal patients ("n" was not given-see below) they examined?

We agree with the reviewer that this is a pivotal limitation of our study and added this as one limitation in the discussion. Our assumption is that activating CBMCs with M-CSF and either IFN- γ or IL-10 mimics a state of inflammation/infection in newborns and that high release of S100A8/9 dampens glycolytic activity and cytokine release. We did not measure S100 levels in cord blood, however it is known that low S100 levels at birth are linked to a higher sepsis risk (Ulas et al. Nat. Immunol. 2017). In detail, amounts of S100A8/A9 greater than 2.000 ng/ml in cord blood were associated with a 25-fold lower risk of LONS compared with levels less than 330 ng/ml. This might on the one side be related to the direct bactericidal capacity of S100A8/A9 proteins, which thus prevents bacterial sepsis. Genetic deletion of S100A8 in mice dramatically enhances susceptibility towards a Staph aureus induced infection model in neonatal mice, which is accompanied by severe hyperinflammation. Our data and in particular the ECAR data (Fig. 6) point to the fact, that S100A8/A9 reduce glycolytic capacity which prevents hyperinflammation (lines 70-74, 186-211, 252-277).

4. It is unclear how many newborns were included in the analyses. E.g. if this data was derived from <10 infants, do the authors suggest the findings could be generalized? Other demographic variables would also be informative (sex, race, etc).

Due to our ethic rules all cord blood samples are anonymized. All experiments were performed with samples from at least 3 different cord bloods unless indicated different.

All term neonates were delivered spontaneously and did not exhibit signs of infection, as defined by clinical status, white blood cell count and C-reactive protein. Mothers with amnion infection and prolonged labour were excluded.

5. Several figures are incomplete or have abbreviations that are not spelled-out in the legend.

Fig 3 is an example. Fig 4. Where are letters B/C?

7. Phagocytic indices are described in the results as being measured but they are not described in methods. *E. coli* incubation (ATCC strain?, CFUs?, duration, media?) also was described in results, and not in methods.

Changed accordingly and added in the methods section: Phagocytosis is a process encompassing multiple steps like binding, uptake and digestion. To dissect these steps, we utilized *E. coli* DH5 α , an encapsulated K12 laboratory strain, carrying the green fluorescent protein (*gfp*)-mut2 gene (*E. coli*-GFP) which was a generous gift from Prof. Dr. Dehio (University of Basel, Switzerland) and was used for phagocytosis as previously described. Bacteria were freshly grown in Lennox-L-Broth-medium (Invitrogen) until early logarithmic growth, resuspended in PBS and used immediately. Infection was performed at a multiplicity of infection (MOI) of 25:1 which was achieved by dilution with PBS. The phagocytosis assays were performed as described. Although this strain is not of clinical relevance its usage clarified the question that no differences in phagocytic activity exists in our experimental setup. The phagocytosis index (CD14⁺GFP⁺ M Φ : CD14⁺ M Φ) was analyzed by flow cytometry.

8. The authors suggest that S100A8/A9 expression is the mechanism behind the mTOR-dependent findings but did not measure S100 proteins nor defend why S100 proteins would still be over expressed 4 days after the cells were removed from the cord blood. Were those genes being over-expressed in the transcript?

We measured transcript levels of S100A8/A9 in cord-blood macrophages and found strongly enhanced transcripts in IFN- γ induced cord-blood macrophages, however not in IL-10 derived macrophages (188-193, supplemental figure 2). We do not know why this differences do not appear in M(IL-10) macrophages. One explanation is that we analyzed RNA expression at the end of in vitro culture (day 4) and that this does not reflect the situation during first critical differentiation steps.

It was shown before that monocytes from newborns produce massive amounts of S100A9 proteins (Ulas et al. Nat. Immunol, 2017, Austermann et al. , Cell Rep., 2014). We assume that this is also the case *in vitro* during our activation protocol at least during the first hours and days.

Minor

1. "Third world" is inappropriate. It would be better to refer to these areas as low-middle income countries (LMIC) or "under resourced". Changed accordingly

2. There is a sentence "...glycolysis, although less efficient in generating ATP, ..." that is repeated 3 times nearly verbatim in the paper. Changed accordingly

We again thank the reviewers and hope that the manuscript will be suitable for publication in your esteemed journal.

Yours sincerely,

Prof. Dr. med. Thorsten Orlikowsky

Reviewers' comments:

Reviewer #2 (Remarks to the Author):

The authors addressed most of my concerns.

Two issues remain:

- The fact that total mTOR is already lower in CB Mph changes the message somehow. There is no deficiency at the level of mTOR activation in CB Mph. There is probably a transcriptional defect of mTOR in CB Mph. This should be tested.
- The authors mention in the Results section that discrepancies were found between Rapamycin treated adult Mph and cord blood Mph. Please specify and discuss these discrepancies.

Reviewer #3 (Remarks to the Author):

The author's responses are appreciated.

Reviewer #2 (Remarks to the Author):

The authors addressed most of my concerns.

Two issues remain:

- The fact that total mTOR is already lower in CB Mph changes the message somehow. There is no deficiency at the level of mTOR activation in CB Mph. There is probably a transcriptional defect of mTOR in CB Mph. This should be tested.

We performed qRT-PCR analysis to determine transcriptional mTOR level of cord blood derived M(IFN- γ) and M(IL-10) macrophages in comparison to adult M(IFN- γ) and M(IL-10) macrophages. MTOR mRNA expression was not reduced in CBMs but even tendentially (not significantly) enhanced. We therefore suggest that other posttranscriptional effects beyond phosphorylation lead to reduced mTOR activation in cord blood derived macrophages. Data are added in Supp. Fig. 1A in the revised manuscript.

- The authors mention in the Results section that discrepancies were found between Rapamycin treated adult Mph and cord blood Mph. Please specify and discuss these discrepancies.

To specify these discrepancies we performed KEGG pathway analysis with M(IL-10) ϕ and M(IFN- γ) from cord blood compared to adult M(IL-10) ϕ / M(IFN- γ) and from Rapamycin treated adult M(IL-10) ϕ / M(IFN- γ) compared to non-Rapamycin treated adult M(IL-10) ϕ / M(IFN- γ) (Supplementary Table 1 and 2). We described this in more detail in the text and also mentioned that other pathways beyond mTOR might critically regulate cord blood derived macrophages.

REVIEWERS' COMMENTS:

Reviewer #2 (Remarks to the Author):

The authors addressed my concerns

Reviewer #3 (Remarks to the Author):

Thank you to the author's for their responses.